# N-Acetyl Serotonin Alleviates Oxidative Damage by Activating Nuclear Factor Erythroid 2-Related Factor 2 Signaling in Porcine Enterocytes

**DOI:** 10.3390/antiox9040303

**Published:** 2020-04-07

**Authors:** Haiwei Liang, Ning Liu, Renjie Wang, Yunchang Zhang, Jingqing Chen, Zhaolai Dai, Ying Yang, Guoyao Wu, Zhenlong Wu

**Affiliations:** 1State Key Laboratory of Animal Nutrition, College of Animal Science and Technology, China Agricultural University, Beijing 100193, China; haiweiliang@cau.edu.cn (H.L.); wangrenjie@cau.edu.cn (R.W.); b20173040318@cau.edu.cn (Y.Z.); B20163040291@cau.edu.cn (J.C.); daizhaolai@cau.edu.cn (Z.D.); cauvet@cau.edu.cn (Y.Y.); 2Institute of Medicinal Biotechnology, Chinese Academy of Medical Sciences and Peking Union Medical College, Beijing 100050, China; dadaliu@cau.edu.cn; 3Department of Animal Science, Texas A&M University, College Station, TX 77843, USA; g-wu@tamu.edu; 4Beijing Advanced Innovation Center for Food Nutrition and Human Health, China Agricultural University, Beijing 100193, China

**Keywords:** N-acetyl serotonin, 4-hydroxy-2-nonenal, nuclear factor erythroid 2-related factor 2, glutathione, intestinal epithelial cells

## Abstract

Apoptosis of intestinal epithelial cells following oxidative stress is a major cause of mucosal barrier dysfunction and is associated with the pathogenesis of various gastrointestinal diseases. Although L-tryptophan (Trp) is known to improve intestinal integrity and function, a beneficial effect of N-acetyl serotonin (NAS), a metabolite of Trp, on the apoptosis of enterocytes and the underlying mechanisms remain largely unknown. In the present study, we showed that porcine enterocytes treated with 4-hydroxy-2-nonenal (4-HNE), a metabolite of lipid peroxidation, led to upregulation of apoptotic proteins, including Bax and cleaved caspase-3, and reduction of tight junction proteins. These effects of 4-HNE were significantly abrogated by NAS. In addition, NAS reduced ROS accumulation while increasing the intracellular concentration of glutathione (GSH), and the abundance of the Nrf2 protein in the nucleus and its downstream target proteins. Importantly, these protective effects of NAS were abrogated by Atra, an inhibitor of Nrf2, indicating a dependence on Nrf2 signaling. Taken together, we demonstrated that NAS attenuated oxidative stress-induced cellular injury in porcine enterocytes by regulating Nrf2 signaling. These findings provide new insights into a functional role of NAS in maintaining intestinal homeostasis.

## 1. Introduction

The intestinal monolayer epithelium serves as the major site for the digestion and absorption of nutrients, electrolytes, and water from the intestinal lumen, while preventing the permeation of toxins, allergens, and pathogens from the gut lumen into mucosal tissue [1]. The enterocytes are vulnerable to oxidative damage due to their constant exposure to reactive oxygen species (ROS) generated from the luminal contents [2]. Oxidative stress, a state of imbalance between the generation of ROS and antioxidant defenses [3,4], has been reported to induce cell apoptosis in both in vitro and in vivo experiments, and therefore contributes to impaired intestinal mucosal barrier function [5,6]. Accumulating evidence shows that oxidative damage in the small intestinal epithelium is a major cause of gut dysfunction and plays a crucial role in the pathogenesis of various gastrointestinal diseases, such as enterocolitis, gastrointestinal cancers, celiac disease, and inflammatory bowel disease [7,8].

4-Hydroxy-2-nonenal (4-HNE), a metabolite of lipid peroxidation in response to intracellular damage, is associated with oxidative stress in various cells and tissues [9]. It has been reported that 4-HNE can regulate B-cell lymphoma 2 (Bcl-2) family proteins, including both pro-apoptotic (Bax, Bak, Bad) and anti-apoptotic (Bcl-2, Bcl-XL) proteins [10]. Activation of pro-apoptotic proteins leads to the release of cytochrome c from the mitochondria to the cytoplasm, which in turn, activates downstream targets of caspases, leading to the morphological and biochemical changes related to apoptosis [11,12,13]. A recent study has shown that the nuclear translocation of nuclear factor erythroid 2-related factor 2 (Nrf2) signaling acts as a critical transcriptional factor associated with cell survival in response to oxidative stress [14]. However, it remains elusive whether Nrf2 signaling is regulated by 4-HNE and therefore contributes to apoptosis in intestinal porcine epithelial cells.

N-acetyl serotonin (NAS), a metabolite of L-tryptophan (Trp), is a precursor for the synthesis of melatonin [15]. Plasma concentration of NAS is ten-fold higher than that of melatonin in human [16]. Moreover, in vitro studies have shown that NAS has a more potent antioxidant capacity than melatonin [15]. NAS administration protects ischemia-reperfusion-induced mucosal damage in the jejunum and ileum of rats [17]. About 25% of dietary Trp is degraded by the porcine small intestine during its first pass into the portal vein and leads to the generation of multiple metabolites including serotonin, melatonin, and NAS [18]. In our recent study, we found that Trp supplementation to the basal diet of weanling piglets improved growth performance and the mucosal barrier function. Further studies revealed that this effect of Trp was associated with enhanced production of tryptamine, a substrate for NAS synthesis in the intestine [19]. At present, the molecular mechanisms responsible for this beneficial effect of Trp are largely unknown. We hypothesized that NAS attenuated 4-HNE-induced oxidative damage and intestinal dysfunction by regulating Nrf2 signaling and contributed to mucosal barrier function. This hypothesis was tested with an intestinal porcine epithelial cell line (IPEC-1), which was originally isolated from the jejunum of an unsuckling piglet [20,21]. This cell line has been validated to be used as a useful cell model for studying intestinal epithelial integrity and barrier function [22,23]. 

## 2. Materials and Methods 

### 2.1. Reagents

NAS was purchased from Sigma Chemical Co. (St. Louis, MO, USA). 4-HNE was bought from Cayman Chemical Co. (Ann Arbor, MI). DMEM/F12 and fetal bovine serum (FBS) were obtained from GIBCO BRL (Grand Island, NY). Antibodies against Nrf2, Bax, Bcl-2 and GAPDH were obtained from Santa Cruz Biotechnology (Santa Cruz, CA). Primary antibodies against tight junction proteins, including zonula occluden (ZO)-1, claudin-1, and occludin were purchased from Invitrogen. Antibodies against cleaved-caspase-3, glutathione synthetase (GSS), heme oxygenase-1 (HO-1), and glutamate-cysteine ligase catalytic subunit (GCLC) as well as the specific antibodies against NAD(P)H quinine oxidoreductase-1 (NQO-1) were purchased from Cell Signaling Technology (Beverly, MA). Peroxidase-conjugated goat anti-rabbit and goat anti-mouse secondary antibodies were purchased from Huaxingbio Biotechnology Co. (Beijing, China). The annexin V-FITC&PI kit was from Jiamay Biotechnology (Beijing, China). All other reagents used in this study were ordered from Sigma.

### 2.2. Cell Culture and Treatment

IPEC-1 cells were cultured as previously described [24]. Briefly, the IPEC-1 cells were grown in DMEM-F12 medium supplemented with 10% FBS and 1% penicillin-streptomycin (Gibco, NY, USA) and were incubated at 37 °C in 5% CO_2_. IPEC-1 cells were pre-treated with NAS (0-250 μM, 12 h) and then were incubated for another 2 h in the absence (served as control) or presence of 4-HNE (20 μM) to induce oxidative damage as has been previously described [20,21]. The doses of NAS used in the present study were based on published studies [14]. 

### 2.3. Cell Viability

Cells (10,000/well) were seeded in a 96-well plate in DMEM-F12 supplemented with 10% FBS to allow for cell adherence, and then were pretreated by different concentrations of NAS (0 μM, 50 μM, 100 μM, and 250 μM) for 12 h, followed by the treatment of 4-HNE (20 μM). Cell viability was assessed by using a cell-counting kit (Zoman Biotech) according to the manufacturer’s instructions. The absorbance of the assay solution was determined at 450 nm by using the microplate reader (Molecular Devices, CA, USA). Results are expressed as a percentage relative to that of controls.

### 2.4. Measurement of Intracellular Reactive Oxygen Species Level

The levels of intracellular ROS were measured by using DCFH-DA (2’,7’-dichlorofluorescein diacetate) as previously described [25]. Briefly, after 2 h incubation of cells with 4-HNE in the presence or absence of NAS, the cells were stained with 10 μM DCFDA and incubated at 37 °C for another 20 min. The intracellular ROS levels were determined by a fluorescence microscope (Zeiss Axiowert) at an excitation wavelength of 485 nm and an emission wavelength of 525 nm, respectively.

### 2.5. Measurement of Intracellular Glutathione

The intracellular glutathione (GSH) concentrations were measured by following the previous method [26]. Briefly, cells were collected and were washed with PBS twice, and then suspended in PBS buffer containing 1 mM EDTA. The mixtures were then vortexed and centrifuged at 12,000 rpm for 15 min at 4 °C. Supernatants separated from cell homogenates were analyzed for GSH level using a Quanticrom Glutathione Assay Kit obtained from BioAssay Systems (Hayward, CA, USA) according to the manufacturer’s protocol.

### 2.6. Apoptotic Cell Death Assay

Cells were treated as indicated and apoptotic cell death was determined by Hoechst 33342/PI staining and flow cytometric analysis according to the previous method [20]. Briefly, IPEC-1 cells preincubated with or without NAS were subjected to 4-HNE treatment. Cells were then harvested and washed with 1× PBS twice, and then were stained with Annexin V-FITC and PI for 15 min at room temperature. Stained cells were analyzed by a flow cytometer (BD Biosciences) with the Cell Quest analysis software.

### 2.7. Extraction of Proteins and Western Blot Analysis

IPEC-1 cells were exposed to 20 μM 4-HNE in the absence or presence of NAS. Cells were lysed on ice for 30 min in RIPA buffer (10 mmol/L Tris-HCl, pH 7.4; 150 nmol/L NaCl; 10 mmol/L EDTA; 1% NP-40; 0.1% SDS) containing protease and phosphatase inhibitors, followed by 12,000 rpm centrifugation for 15 min at 4 °C. The supernatant fluid was collected. The proteins were extracted from the nucleus and the cytosol of cells were conducted by using a kit from the Beyotime Institute of Biotechnology (Haimen, China). Protein concentration was determined by the Pierce BCA protein Assay Kit (Huaxingbio, Beijing). Samples of total protein lysates (25 μg/sample) were injected into 10% SDS-PAGE gels and transferred onto PVDF membranes (Millipore, Billerica, MA). The membranes were blocked with 5% skimmed-milk solution in Tris-buffered saline containing 0.05% Tween-20 (TBST) for 30 min at 25 °C, and then were incubated with one of the indicated primary antibodies (1: 2000) overnight at 4 °C. Thereafter, the membranes were washed with TBST three times and then were incubated with the horseradish peroxidase (HRP)-conjugated secondary antibody (1:5000) for 1 h. The protein bands were developed by an enhanced chemiluminescence kit (Applygen Technologies Inc., Beijing, China) using the ImageQuant LAS 4000 mini system (GE Healthcare). Quantification of band density was performed by using the Quantity One software (Bio-Rad Laboratories).

### 2.8. Immunofluorescence Microscopy

IPEC-1 cells fixed with 4% paraformaldehyde were blocked with 10% BSA and then incubated with a primary antibody (1:300) against ZO-1 overnight at 4 °C. The cells were incubated with an appropriate secondary antibody (1:500) at 25 °C for 1 h. Nuclei were stained with the use of Hoechst 33258 (1 mg/mL). Distribution of ZO-1 protein was visualized under a fluorescence light microscope (Axio Vert.A1, Zeiss, Oberkochen, Germany).

### 2.9. Statistical Analysis

Experimental data on apoptosis, the relative protein grayscale and the relative gene expression levels were expressed as means ± SEM and were analyzed by one-way ANOVA and the Duncan multiple comparison method with the use of the SPSS statistical software (SPSS, Inc., Chicago, IL, USA). *P*-values < 0.05 were taken to indicate statistical significance.

## 3. Results

### 3.1. NAS Attenuated the 4-HNE-Induced Apoptosis of IPEC-1 Cells 

Compared with the control, cells treated with 4-HNE had a reduced cell viability, which was significantly attenuated by NAS in a dose-dependent manner (Figure 1A). Among the doses of NAS used in our study, cells treated with 100 μM of NAS had greatest protection as shown by enhanced cell viability, compared with cells exposed to 4-HNE. Treatment with NAS alone had no effect on the viability of IPEC-1 cells (Figure 1A). Western blot analysis showed that 4-HNE treatment led to enhancement of the protein level of cleaved caspase-3, a characteristic of apoptosis, which was abrogated by 100 μM of NAS (Figure 1B and 1C). The protein level of pro-apoptotic Bax was upregulated by 4-HNE, however, was not affected by the presence of NAS. The protein level of Bcl-2 was not affected by 4-HNE single treatment. In contrast, anti-apoptotic protein Bcl-2 was markedly increased by the 4-HNE plus NAS co-treatment (Figure 1B and 1C). The flow cytometric analysis indicated that 4-HNE-induced cell death was significant alleviated by NAS (50-100 μM), with 100 μM of NAS showing a greater protective effect (Figure 1D and 1E). These results suggested that NAS administration attenuated the 4-HNE-induced apoptosis of IPEC-1 cells.

### 3.2. The Effect of NAS on ROS and Intracellular GSH Concentrations 

To investigate the involvement of ROS in the death of 4-HNE-hallenged cells and its contribution to cell death, intracellular ROS levels were determined in the present study. Compared with the controls, 4-HNE treatment led to increased intracellular ROS levels as shown by the DCFH-DA probe, a classic method to detect the generation of ROS by fluorescence staining. Addition of NAS to cell culture reduced the intracellular levels of ROS in a dose-dependent manner (Figure 2A). The Quanticrom Glutathione Assay was used to determine intracellular GSH concentrations. We found that cells treated with 4-HNE had a decreased (*P* < 0.05) level of GSH, as compared with the control (Figure 2B), and this effect of 4-HNE was reversed by NAS administration. Moreover, morphological observation using a phase contrast microscopy demonstrated that 4-HNE treatment increased the number of floating cells, the appearance of cell shrinkage and boundary contraction, as compared with the control cells, and these effects of the oxidant were reversed by NAS supplementation (Appendix A).

### 3.3. NAS Enhanced the Abundance of Tight Junction Proteins in 4-HNE Challenged Cells 

Intestinal epithelial cells were tightly bound together by tight junction proteins. A proper function of tight junction proteins was critical for the maintenance of the intracellular homeostasis and mucosal barrier function [27]. In consistency with the observed phenotypes, incubation of IPEC-1 cells with 4-HNE caused a significant decline in the protein abundances of ZO-1, occludin, and claudin-1, as compared with the control group (*P* < 0.05). This effect of 4-HNE was prevented by NAS supplementation (Figure 3).

### 3.4. NAS Regulated the Nrf2 Signaling in IPEC-1 

As shown in Figure 4, 4-HNE treatment decreased the protein level of Nrf2 in the nucleus (Figure 4A), without affecting the total protein level of Nrf2 (Figure 4B), indicating inactivation of Nrf2 signaling in response to oxidative stress. This effect of 4-HNE was reversed by 100 μM of NAS, as evidenced by the enhanced protein level of Nrf2 in the nuclear compartment (Figure 4A). These results indicated the regulatory effect of NAS on Nrf2 signaling and its potential contribution to the viability of enterocytes. Considering that 100 μM of NAS had a greater protective effect than other doses used in the present study, this concentration was used in the following experiment for the mechanistic study.

### 3.5. NAS Protected Cells Against 4-HNE-Induced Apoptosis in a Nrf2-Dependent Manner 

To validate the functional role of Nrf2 on 4-HNE-induced apoptosis, cells pretreated with or without Atra, a specific inhibitor of the Nrf2, were incubated with 4-HNE in the presence or absence of NAS. Compared with the control, 4-HNE treatment led to reduced cell viability, which was abolished by NAS administration (Figure 5A). Western blot analysis showed that 4-HNE treatment led to the decreased protein level of Nrf2 in the nucleus, and its downstream targets, HO-1 and NQO-1, as well as reduced protein levels of Bcl-2, GCLC, and GSS, and these effects of 4-HNE were abrogated by NAS (Figure 5B, and Appendix A). Importantly, the regulatory effect of NAS on the Nrf2 signaling proteins were reversed by Atra. The protein level of Bax was upregulated by 4-HNE, however, was not affected by either NAS plus 4-HNE co-treatment or NAS+4-HNE+Atra combination treatment. Of note, the protein levels of Bax were lowered by NAS or Atra single treatment. Moreover, we also found that the alleviated effect of NAS on 4-HNE-induced ROS accumulation was abolished by Atra (Figure 5C). These data indicated that Nrf2 was responsible for the protective effect of NAS on 4-HNE-induced apoptosis in intestinal epithelial cells.

### 3.6. NAS Restored Tight Junction Proteins in 4-HNE-Treated Cells by Regulating Nrf2 Signaling 

To assess an implication of Nrf2 signaling and its contribution to tight junction protein regulation, the proteins abundances of ZO-1, claudin-1, and occludin were determined in 4-HNE challenged cells in the presence or absence of Atra. As shown in Figure 6, 4-HNE single treatment resulted in the reduced protein level of ZO-1, claudin-1, and occludin, and these effects of 4-HNE were reversed by NAS (Figure 6A and 6B). Intriguingly, such effects of NAS on tight junction proteins in the enterocytes were abolished by Atra, an inhibitor of Nrf2 in the enterocytes. Immunofluorescence staining was performed to analyze the location of tight junction proteins. As shown in Figure 6C, NAS administration increased the abundances of ZO-1 localized between neighboring cells, and this effect of NAS was attenuated by Atra.

## 4. Discussion

The balance between ROS generation and the antioxidant defense systems is crucial to intestinal homeostasis [7]. Accumulation of ROS is associated with reduced intestinal mucosal barrier, increased intestinal permeability, as well as the development of various intestinal diseases, such as enterocolitis, gastrointestinal cancers, celiac disease, and inflammatory bowel disease [7,8,28]. As a metabolite of Trp, NAS has a strong antioxidant capacity [15]. It has been reported that plasma concentrations of NAS are at nanomolar levels, which are greater than those of melatonin [16]. Moreover, NAS has been approved by the FDA for treatment of neurological disorders and stroke [29]. Recent studies have shown that NAS protects against oxidative damage in neurons, erythrocytes, retinal cells, lung epithelial cells, lymphocytes, and enterocytes [17,29,30,31,32]. However, the underlying mechanisms responsible for the beneficial effect of NAS are largely unknown. In the present study, we found that NAS administration attenuated the 4-HNE-induced ROS accumulation and cell death, while enhancing the abundance of tight junction proteins. Further experiments showed that activation of Nrf2 signaling was critical for the protective effect of NAS against oxidative stress.

4-HNE has been reported to be associated with intestinal epithelial cell injury by triggering apoptosis in enterocytes [33]. We noted that 4-HNE treatment led to 25% cell death, which was less than that reported in a previous study [33]. Toxicity of 4-HNE is highly related to the dosage used, exposure time, and cell density. The concentration of 4-HNE (20 μM) used in the present study was lower than that of the previous study (40 μM) [33], therefore leading to reduced apoptosis. To determine the antioxidant property of NAS against 4-HNE-induced oxidative damage, IPEC-1 cells were treated with 4-HNE in the presence of different concentrations of NAS. In agreement with our previous study [33], 4-HNE treatment led to caspase-3 dependent apoptosis, as shown by the Facs analysis and Western blot results. A novel and important finding of this work is that the 4-HNE-induced upregulation of Bax and cleavage of caspase-3 was significantly abrogated by NAS administration. This effect of NAS, along with the upregulation of a key anti-apoptotic factor Bcl-2, contributed to the survival of 4-HNE-challenged cells. Among the doses of NAS used in the present study, 100 μM of NAS showed greatest protection as compared with other doses.

To further explore the involvement of ROS in apoptosis of the enterocytes, intracellular ROS was determined by the DCFH-DA probe. The 4-HNE-induced ROS accumulation was markedly reduced by NAS. Intracellular GSH plays an important role in antioxidant defense, which is essential for cell survival and elimination of intracellular ROS in the intestinal epithelium [34]. Depletion of intracellular GSH is associated with increased oxidative stress and increased risks for numerous chronic diseases [35]. As expected, 4-HNE-induced GSH depletion was substantially reversed by NAS administration. The de novo synthesis of GSH is critical for the scavenging of ROS and maintenance of redox status. NAS supplementation enhanced GSH synthesis, thereby contributing to the restoration of intracellular homeostasis in response to 4-HNE. These results support an anti-oxidative capacity of NAS and its contribution to improving cell survival in enterocytes [26,36]. Considering that NAS is a precursor for melatonin synthesis, further studies are needed to elucidate whether the beneficial effect of NAS is predominantly due to the presence of the antioxidant or its metabolites melatonin.

Nrf2 signaling is a transcriptional factor associated with cellular survival by regulating the expression of genes involved in GSH biosynthesis and antioxidative defense, such as GCLC, GSS, NQO1, and HO-1 [26,37]. Disruption of Nrf2 signaling is associated with an increased susceptibility to oxidative insults in various conditions [38,39]. Therefore, activation of Nrf2 signaling has been regarded as a promising strategy to combat oxidative stress-related injury [40]. In our study, we found that NAS treatment led to an enhanced abundance of GSS, a key enzyme involved in GSH synthesis, boosted the protein level of Nrf2 in the nucleus fragment and its downstream targets, HO-1 and NQO1, two detoxifying enzymes participated in the scavenging of ROS [41,42], thus restoring the intracellular redox and protecting intestinal epithelial cells from 4-HNE-induced cell damage. Importantly, these effects of NAS were abolished by Atra, a specific inhibitor of Nrf2, indicating a dependence on Nrf2 signaling. In consistency with the phenotypes, a regulatory effect of NAS on tight junction proteins (claudin-1, occludin, and ZO-1), as shown by both the Western blot result and immunofluorescence staining, was abolished by a Nrf2 inhibitor. These data validated a critical role of Nrf2 signaling in the survival of enterocytes and intestinal mucosal barrier function in response to oxidative damage. NAS may be a useful additive to the diets of stress-challenged animals (including weanling pigs), which generally exhibit reductions in growth, feed efficiency, and immunity [43], for enhancing their health and productivity.

## 5. Conclusions

Results from the present study indicated that NAS administration protected enterocytes against oxidant-induced cell death and intestinal barrier dysfunction. Specifically, NAS regulates intracellular redox states by upregulation of enzymes involved in GSH biosynthesis, enhancing abundance of proteins involved in anti-oxidative defense, while decreasing apoptotic proteins in a Nrf2-dependent manner. Considering that NAS had no effect on cell viability of enterocytes, however, ameliorated cellular damage and improved intestinal barrier function in response to oxidative stress, the supplementation of NAS might be a potential strategy to prevent intestinal disorders caused by oxidative insults in humans and animals.

## Figures and Tables

**Figure 1 antioxidants-09-00303-f001:**
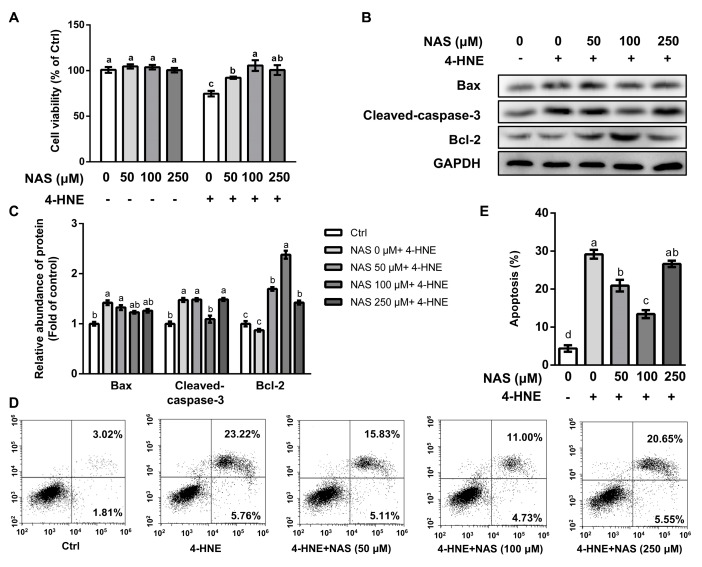
N-acetyl serotonin (NAS) attenuated the 4-HNE-induced apoptosis of IPEC-1 cells. IPEC-1 cells pre-treated with NAS (0- 250 μM, 12 h) were treated with or without 4-HNE (20 μM) for 2 h. Cell viability (**A**), the protein abundances of Bax, cleaved caspase-3, Bcl-2, and GAPDH (**B**), protein band density (**C**), apoptosis (**D**), and its statistical analysis (**E**) are shown. Representative results from three independent experiments were provided, and values are means ± SEMs, n = 3. Means without a common letter differ, *p*< 0.05.

**Figure 2 antioxidants-09-00303-f002:**
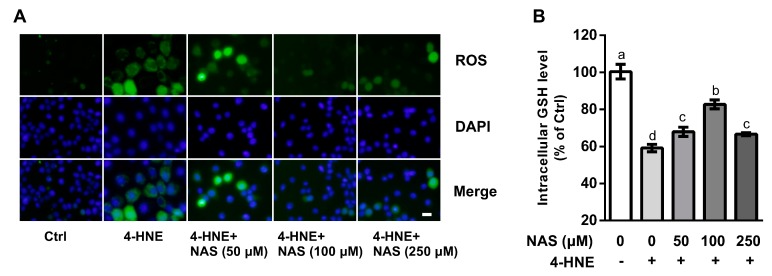
Effects of NAS on reactive oxygen species (ROS) and intracellular glutathione (GSH). Cells were left untreated or treated with 4-HNE in the absence or presence of NAS administration as indicated. (**A**) ROS were determined and visualized by a fluorescence microscope, magnification ×100. (**B**) intracellular GSH concentrations were measured. Representative results from three independent experiments were provided, and values are means ± SEMs, n = 3. Means without a common letter differ, *P* < 0.05.

**Figure 3 antioxidants-09-00303-f003:**
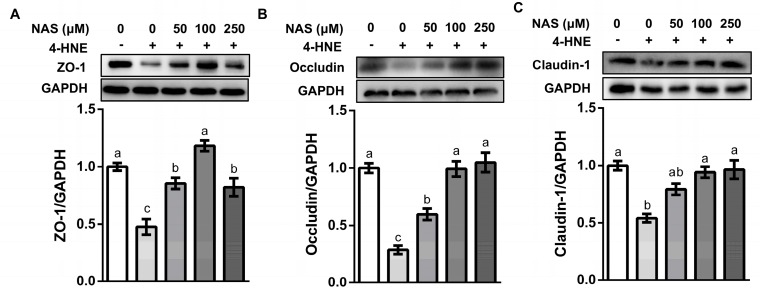
NAS enhanced the abundance of tight junction proteins. IPEC-1 cells were treated as described in Figure 2. Protein abundance for ZO-1 (**A**), occludin (**B**), and claudin-1 (**C**) were determined by Western blot analysis. Representative results from three independent experiments were provided, and values are means ± SEMs, n = 3. Means without a common letter differ, *P* < 0.05.

**Figure 4 antioxidants-09-00303-f004:**
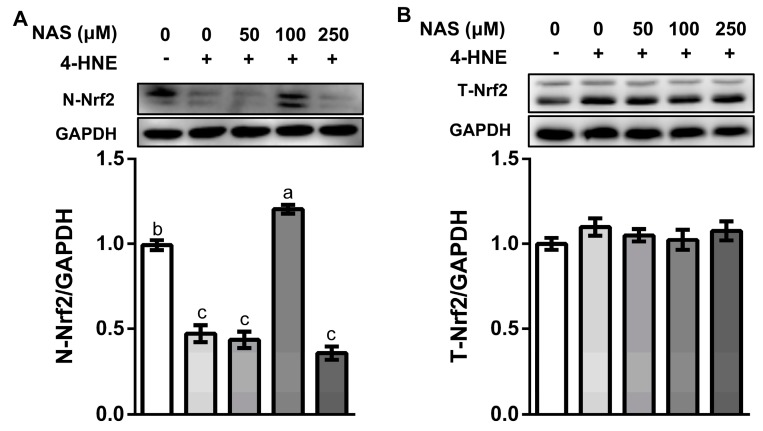
NAS regulated Nrf2 signaling. IPEC-1 cells were treated as described in Figure 2. Protein abundances for nucleus Nrf2 (**A**) and total Nrf2 (**B**) were determined by Western blot analysis. Representative results from three independent experiments were provided, and values are means ± SEMs, n = 3. Means without a common letter differ, *P* < 0.05.

**Figure 5 antioxidants-09-00303-f005:**
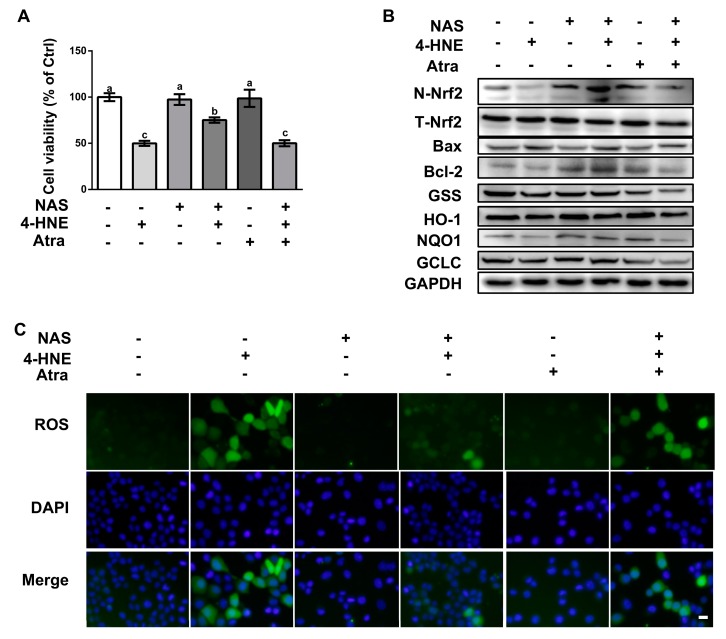
NAS protected cells against 4-HNE-induced apoptosis in a Nrf2-dependent manner. Cells pretreated with or without NAS (100 μM, 12 h) were exposed to 4-HNE (20 μM) for 2 h in the presence or absence of Atra (2 μM). Cell viability (**A**), the protein abundances for N-Nrf2, T-Nrf2, Bax, Bcl-2, GSS, HO-1, NQO1, and GCLC (**B**), and intracellular ROS levels (**C**) were assessed. Representative results from three independent experiments were provided, and values are means ± SEMs, n = 3. Means without a common letter differ, *P* < 0.05.

**Figure 6 antioxidants-09-00303-f006:**
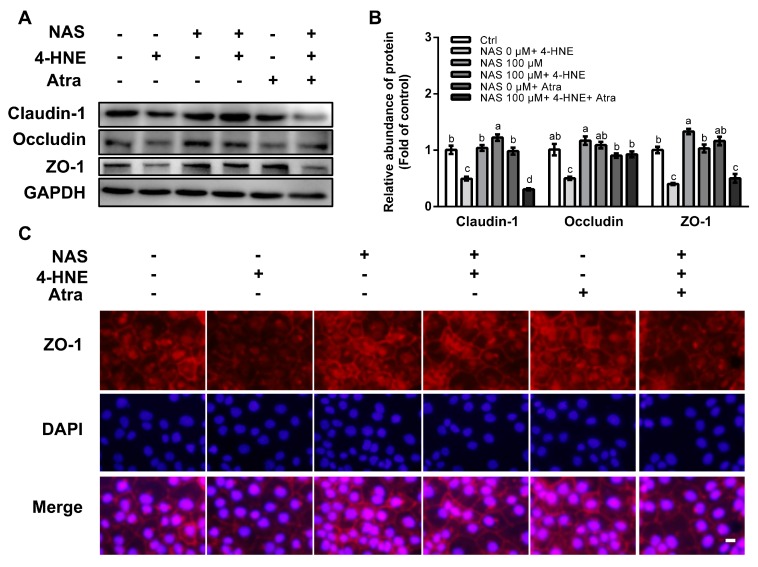
NAS regulated tight junction proteins in Nrf2-dependent manner. Cells were treated as described in Figure 5. Protein abundances of claudin-1, occludin, and ZO-1 (**A**), protein band density (**B**), and immunofluorescence staining for ZO-1 (**C**) were determined. Representative results from three independent experiments were provided, and values are means ± SEMs, n = 3. Means without a common letter differ, *P* < 0.05, magnification ×100.

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
