# Peer review of "N-Acetyl Serotonin Alleviates Oxidative Damage by Activating Nuclear Factor Erythroid 2-Related Factor 2 Signaling in Porcine Enterocytes"

_antioxidants, 2020, doi:10.3390/antiox9040303_

Round 1

Reviewer 1 Report

General comments

The authors studied mechanisms on how N-acetyl serotonin (NAS) could protect epithelial cells from oxidative stress. Nrf2 is a well-known transcriptional factor, and the signaling is a representative of antioxidative defense systems. The authors showed that the protective effect against 4-hydroxy-nonenal (4-HNE) of NAS also was via the Nrf2 signaling. This research should fit the scope of “Antioxidants.” However, the group of Dr. Wu had already published similar data (reference No. 30) using the same epithelial cells with 4-HNE and N-acetylcysteine (NAC). The authors should explain the significance of this study well to convince readers. Otherwise, this study looks a mimic of reference No. 30, where NAC was used instead of NAS. Does the hypothesis proposed in the reference No. 30 have to be independent of the findings produced with NAS in this study? Unless the authors discuss enough, including their previous data, the scientific significance of this paper sounds just like that NAS was used instead of NAC.

Major comments

  1. Lines 59-61. The authors must explain the reason why the porcine intestinal epithelial cell line was used in this study. If the authors wanted to evaluate the effect of NAS on human cells, Caco-2, HT-29, and T84 cells had to be served as the in vitro models of small and large intestinal epithelia, respectively.
  2. 1E. Six hours incubation with 4-HNE (20 μM) caused apoptosis only in 23-30% of IPEC-1 cells. This result suggests that 4-HNE is less toxic on the IPEC-1 cells compared to published data of other cells. This reviewer would like to request the authors to describe a discussion about this phenomenon because the properties may affect the universality of this model.  
  3. Figure 4. Three to five experiments were done, and values are means (n=3). There is no description of the statistical method of how three data were chosen among the data from five experiments.
  4. Lines 203-205. This reviewer cannot see the apparent change in the HO-1 level among treated samples.
  5. Line 218. If the authors would like to refer to integrity, this reviewer recommends that the authors should show the transepithelial electric resistance (TEER).

Minor comments

  1. Lines 19 and 42. “4-hydoxy-2-nonenal” should be “4-hydroxy-2-nonenal”
  2. Line 38. Reference 7 is for osteoblasts and seems to be inadequate in the sentence context.
  3. Lines 45-48. The authors referred to Bcl-2 as anti-apoptotic on line 45. However, Bcl-2 was introduced as apoptotic on lines 46-48. The discrepancy should be solved.
  4. Line 244. A brief explanation of the mechanism on how 4-HNE induced ROS accumulation should be kindly given so that readers can understand. A schematic figure showing the proposed mechanism of how 4-HNE and NAS act in this epithelial model would be convenient for readers, learning the Figure 7F of Reference 30.

Author Response

Point 1: The authors studied mechanisms on how N-acetyl serotonin (NAS) could protect epithelial cells from oxidative stress. Nrf2 is a well-known transcriptional factor, and the signaling is a representative of antioxidative defense systems. The authors showed that the protective effect against 4-hydroxy-nonenal (4-HNE) of NAS also was via the Nrf2 signaling. This research should fit the scope of “Antioxidants.” However, the group of Dr. Wu had already published similar data (reference No. 30) using the same epithelial cells with 4-HNE and N-acetylcysteine (NAC). The authors should explain the significance of this study well to convince readers. Otherwise, this study looks a mimic of reference No. 30, where NAC was used instead of NAS. Does the hypothesis proposed in the reference No. 30 have to be independent of the findings produced with NAS in this study? Unless the authors discuss enough, including their previous data, the scientific significance of this paper sounds just like that NAS was used instead of NAC.

Response:

Thanks for the comments. We agree that both our previous study (reference No. 30) and the present one address the anti-oxidative damage using IPEC-1 cells. In our previous study, NAC was used and its underlying mechanisms was investigated. We found that NAC protected apoptosis by promote GSH synthesis and MKP-1 mediated inhibition on ERK1/2 signalling. Even though anti-oxidative effect of NAC is well known, our study uncovered a underlying mechanisms for this beneficial effect.

     Our lab is mainly focus nutritional biochemistry study with an aim to improved intestinal barrier and health using piglet as a model. In our recent study, we found that tryptophan supplementation to basal diet of weanling piglets improved growth performance and intestinal barrier. Further study shown that this effect of tryptophan was associated with enhanced tryptophan metabolites, such as kynurenine, kynurenic acid, 5- hydroxytryptophan, melatonin, and NAS in the intestine. Based on these in vivo studies, we conducted experiments to test effect of anti-oxidative effect NAS in intestinal porcine intestinal cells (IPEC-1), as shown in this study. We have added more sentences in the introduction section on our research hypothesis of this version. The conclusion of this work provided basis for the regulation mechanism of NAS (tryptophan metabolite) on the intestinal epithelial antioxidation. Thanks!

Point 2: Lines 59-61. The authors must explain the reason why the porcine intestinal epithelial cell line was used in this study. If the authors wanted to evaluate the effect of NAS on human cells, Caco-2, HT-29, and T84 cells had to be served as the in vitro models of small and large intestinal epithelia, respectively.

Response:

Thanks for the suggestions. We agree that Caco-2, HT-29, and T84 cells are widely used for in vitro study on human intestinal epithelial related studies. We selected IPEC-1 as a model is based on our in vivo study in piglet, as above-mentioned (in Point 1). We hope our in vitro study might helpful to address what we found in piglets. So, we did not used the human cells. We have added one sentence to explain this point in the introduction section of this version. Thanks!

Point 3: 1E. Six hours incubation with 4-HNE (20 μM) caused apoptosis only in 23-30% of IPEC-1 cells. This result suggests that 4-HNE is less toxic on the IPEC-1 cells compared to published data of other cells. This reviewer would like to request the authors to describe a discussion about this phenomenon because the properties may affect the universality of this model.

Response:

The toxicity of 4-HNE is highly related to the dosage used, exposure time, and cell density. In the previous study, the cells were treatment with 40 μM of 4-HNE and led to 50% cell death. In the present study, the cells were treated with 20 μM of 4-HNE, and we found 30% cell death following 2 h exposure. We have added sentences in the discussion section of the revised manuscript on this point (P8, L262-265). Thank you very much!

Point 4: Figure 4. Three to five experiments were done, and values are means (n=3). There is no description of the statistical method of how three data were chosen among the data from five experiments.

Response:

Thanks for the comments. We have corrected the sentences in each figure to make it clear in this version. Sorry for the confuse caused.

Point 5: Lines 203-205. This reviewer cannot see the apparent change in the HO-1 level among treated samples.

Response:

Thanks for the comments. In our study, protein level of HO-1 was changed. We have enlarged the figure to make it readable in the version. Moreover, the band densities were provided in this version (supplemental Figure1). Thanks!

Point 6: Line 218. If the authors would like to refer to integrity, this reviewer recommends that the authors should show the transepithelial electric resistance (TEER).

Response:

Thanks for the suggestions. We have corrected the word to make it more accurate and being supported by our data provided herein. Thanks.

Point 7: Lines 19 and 42. “4-hydoxy-2-nonenal” should be “4-hydroxy-2-nonenal”

Response:

Sorry for the error. We have corrected it as suggested.

Point 8: Line 38. Reference 7 is for osteoblasts and seems to be inadequate in the sentence context.

Response:

We have replaced it, thanks!

Point 9: Lines 45-48. The authors referred to Bcl-2 as anti-apoptotic on line 45. However, Bcl-2 was introduced as apoptotic on lines 46-48. The discrepancy should be solved.

Response:

We have corrected it in this version (Page 2, Line 50-51), thanks!

Point 10: Line 244. A brief explanation of the mechanism on how 4-HNE induced ROS accumulation should be kindly given so that readers can understand. A schematic figure showing the proposed mechanism of how 4-HNE and NAS act in this epithelial model would be convenient for readers, learning the Figure 7F of Reference 30.

Response:

Thanks for the helpful suggestions. We have a schematic figure for the mechanism and the journal editor ask us to submitted as a separate file as a graphic abstract later. Thank!

Reviewer 2 Report

This is an interesting manuscript about possible mechanism of NAS effect on enterocytes. However, there are some minor issues that need to be addressed:

  • In the section Material and Methods it should be clearly stated what type of cells were considered control cells and which treatments were compared and why (it is not clear from this section if there was a population of cells that were not treated with NAS)
  • In the same section it could be helpful for discussion and final conclusion to state the reasons for selected NAS concentrations and 4-HNE concentration
  • error bars on all graphs should be in both + and – direction in order to more easily understand differences between cell populations and dispersion in each sample
  • given that all statistical analysis that were considered significant are written just as p <05 it is advisable to perform multiple testing corrections such as Bonferroni correction and state exact p-value
  • the level of intracellular ROS was measured as described for the cyanobacterium – is this method reliable for interpretation on eukaryotic cells as well?
  • whole study was performed on porcine cell line – why not use human cell line if conclusion proposes strategy for prevention of intestinal disorder in humans?
  • having in mind that study is based on the cell line (not epithelial tissue or at least primary cell culture) conclusion for intestinal barrier dysfunction is questionable

Author Response

Point 1: In the section Material and Methods it should be clearly stated what type of cells were considered control cells and which treatments were compared and why (it is not clear from this section if there was a population of cells that were not treated with NAS).

Response:

Thanks for the helpful suggestions. IPEC-1 cells pre-treated with NAS (0- 250 μM, 12 h) were treated with or without 4-HNE (20 μM). That is, IPEC-1 cells that were not treated with any drug were used as control cells, as shown the first sample in Figure 1. Our main research focus in this study was to investigate whether the deleterious effect of 4-HNE was attenuated by NAS administration. So, we also compared cells treated with 4-HNE with those cells treated with 4-HNE and NAS combination in our study. We have modified the sentences in the result section to make it clear in this version. Thanks!

Point 2: In the same section it could be helpful for discussion and final conclusion to state the reasons for selected NAS concentrations and 4-HNE concentration.

Response:

Thanks for the suggestions. The oxidative stress induced by 4-HNE was conducted as the previous studies with slight modification (Liu et al., 2018; Ji et al., 2018) and the selection of NAS concentration was based on the previous study (Yoo et al., 2017). We have modified sentences in the materials and methods section to make it clear in this version (Page 2, Line 79-81).

Point 3: Error bars on all graphs should be in both + and – direction in order to more easily understand differences between cell populations and dispersion in each sample.

Response:

Thank you very much. We have modified the bar graphs in our manuscript to show both the direction of Error bars as suggested. Thanks!

Point 4: Given that all statistical analysis that were considered significant are written just as p <05 it is advisable to perform multiple testing corrections such as Bonferroni correction and state exact p-value.

Response:

Thanks for the suggestions. In our study, data were first analysed by one-way ANOVA, and then the Duncan multiple comparison method were performed with the use of SPSS statistical software (SPSS, Inc., Chicago, IL, USA). This statistical analysis is in agreement with previous study (Su et al., 2017). Thanks.

Point 5: The level of intracellular ROS was measured as described for the cyanobacterium – is this method reliable for interpretation on eukaryotic cells as well?

Response:

Thanks for the comments. The level of intracellular ROS was measured using DCFH-DA (2’,7’-dichlorofluorescein diacetate). DCFH-DA is commonly used to detect the generation of reactive oxygen species (ROS) in many types of cells, especially the endothelial cells, such as human umbilical vein endothelial cells (Song et al., 2017) and human corneal epithelial cells (Park et al., 2018). So, it’s a very important and reliable method for the detection of ROS. We have replaced the reference in this version to make it more accurate in this version (Page 3, Line 95).

Point 6: Whole study was performed on porcine cell line – why not use human cell line if conclusion proposes strategy for prevention of intestinal disorder in humans?

Response:

In our in vivo study using piglet as an animal model, we found that NAS has anti-oxidant capacity. To make our further study more relevant to our initial study, we used intestinal porcine epithelial cells in the present study. Also, this cell lines (IPEC-1) has been widely used for studies on intestinal health and nutrition in lots of studies. Thanks.

Point 7: Having in mind that study is based on the cell line (not epithelial tissue or at least primary cell culture) conclusion for intestinal barrier dysfunction is questionable

Response:

Thanks for your comments. More and more studies have shown that pig is a better animal model considering its similarity in physiology, metabolism, and nutrition, as compared with other animals. IPEC-1 cells used in this study is a cell lines original isolated from jejunum of piglet and has been used for intestinal barrier function related studies (FASEB J. 2020 Feb;34(2):2483-2496; Vet Immunol Immunopathol. 2020 Feb; 220:109989; Biochimie. 2020 Mar; 170:10-20; Nutrients. 2018 May 9;10(5). pii: E588.). We have shown that IPEC-1 cells can form trans-epithelial electronic resistance and translocation of tight junction proteins to the interface of neighbouring cells (J Nutr. 2019 Nov 1;149(11):1904-1910; J Nutr. 2018 Apr 1;148(4):526-534; J Anim Sci Biotechnol. 2017 Aug 16; 8:66. doi: 10.1186/s40104-017-0186-0. eCollection 2017). These finding validating that IPEC-1 cells can be used for nutrition related study. We have tried to isolated primary cell lines for our study and did not make it. Thanks for your suggestion.

Reviewer 3 Report

In this report the authors utilized a cell model (an intestinal porcine epithelial cell line (IPEC-1)) to test the hypothesis that administration of N-acetyl serotonin attenuates intestinal dysfunction and oxidative damage caused by 4-Hydoxy-2-nonenal (4-HNE) via regulation of nuclear factor erythroid 2-related factor 2 (Nrf2) signaling. This hypothesis was supported by the present results suggesting that NAS administration protected enterocytes against oxidant-induced cell death and intestinal barrier dysfunction. The results have practical relevance: the authors concluded that a supplementation of NAS might be a used for preventing intestinal disorder caused by oxidative insults. 

Sometimes English might be improved (e.g., “Moreover, in vitro study has showed that NAS has a potent antioxidant capacity than melatonin [16]”).

Author Response

Point 1: Sometimes English might be improved (e.g., “Moreover, in vitro study has showed that NAS has a potent antioxidant capacity than melatonin [16]”).

Response:

The suggested alterations have made (Page 2, Line 60). Thanks!

Reviewer 4 Report

The manuscript showed that N-acetyl serotonin alleviates oxidative damage and this is from activation of nuclear factor erythroid 2-related factor 2 signaling, and the study used porcine enterocyte cells. It is a bit interesting but the issues mentioned below should be addressed before published in this journal.

  1. In Figure 1.A, the authors claimed that NAS 100uM/4-HNE group shows greatest protection. However, in Figure 1.B, although Bcl-2 protein expression is higher in NAS 100uM/4-HNE treated group than other groups, the authors described “anti-apoptotic protein Bcl-2 was not affected by 4-HNE.” This explains that higher expression of Bcl-2 protein in NAS 100uM/4-HNE group might have influence anti-apoptotic effect.
  2. In Figure 1.B, Caspase-3 protein level should be evaluated to precisely compare the level of cleaved-caspase-3 protein expression among the groups.
  3. In the legend of Figure 2, ‘Cell were treated as in Fig.1.’ should be ‘Cells were treated as in Fig. 1.’ In addition, higher quality pictures are required for Figure 2A, 5C, 6C.
  4. In Figure 5.B, the protein level of Bax, an anti-apoptic marker, in Atra negative groups, was higher in NAS-only treatment group than 4-HNE-only counterpart. This is not matched with the authors’ claims described in Figure 1. In addition, there is no description about GCLC of Figure 5B in the main text.
  5. In Figure 6C, false positive signals of ZO-1, which should be expressed specifically on cell surface, were shown in nuclei and the changes of the signals by NAS, 4-HNE and Atra was also shown remarkably in nuclei when compared with cell surface. Figure should be replaced.
  6. It is interesting study but there is few new finding when compared with previous published papers. For example, <Neuroprotective action of N-acetyl serotonin in oxidative stress-induced apoptosis through the activation of both TrkB/CREB/BDNF pathway and Akt/Nrf2/Antioxidant enzyme in neuronal cells; DOI: 10.1016/j.redox.2016.12.034 >, < N-Acetyl-Serotonin Protects HepG2 Cells from Oxidative Stress Injury Induced by Hydrogen Peroxide, doi.org/10.1155/2014/310504> and others. The reviewer recommends that the authors address their new findings and also change the title of this study.

Author Response

Point 1: In Figure 1.A, the authors claimed that NAS 100uM/4-HNE group shows greatest protection. However, in Figure 1.B, although Bcl-2 protein expression is higher in NAS 100uM/4-HNE treated group than other groups, the authors described “anti-apoptotic protein Bcl-2 was not affected by 4-HNE.” This explains that higher expression of Bcl-2 protein in NAS 100uM/4-HNE group might have influence anti-apoptotic effect.

Response:

Yes, we agree that induction of Bcl-2 is implicated in the protective effect of NAS in our study. We have modified the sentences to make it clear in this version (Page 2, Line 50). Thanks!

Point 2: In Figure 1.B, Caspase-3 protein level should be evaluated to precisely compare the level of cleaved-caspase-3 protein expression among the groups.

Response:

Thanks for the suggestions. In response to 4-HNE treatment, caspase 3 was cleaved to its active form (cleaved-caspase 3), which interacts with other apoptotic proteins and transduce the cell signalling. Therefore, the cleaved-caspase 3 is used as a marker for apoptosis in our study. This is in agreement with previous studies. In some papers, the antibodies used for caspase-3 can detect both procaspase 3 and cleaved-caspase 3, so they got two bands. However, in our study, the antibody is specific for cleaved-caspase 3, so there was only one band as shown in our results. Your suggestion is very reasonable and we are sorry for not amending an additional caspase-3 protein level in this version.

Point 3:

In the legend of Figure 2, ‘Cell were treated as in Fig.1.’ should be ‘Cells were treated as in Fig. 1.’ In addition, higher quality pictures are required for Figure 2A, 5C, 6C.

Response:

We have corrected the sentence in this figure, thanks! Also, we have replaced Figure 2A, 5C, 6C with better ones in this version. Thanks for the helpful suggestions.

Point 4: In Figure 5.B, the protein level of Bax, an anti-apoptic marker, in Atra negative groups, was higher in NAS-only treatment group than 4-HNE-only counterpart. This is not matched with the authors’ claims described in Figure 1. In addition, there is no description about GCLC of Figure 5B in the main text.

Response:

Sorry for this unclear description. In our study, we found that 4-HNE single treatment increased Bax protein, but this upregulation was not affected by NAS plus 4-HNE co-treatment as shown in Fig 1. Similar result was found in Fig5B. We have revised the sentence to make it clear as suggested. Also, we have a statistical analysis for the proteins as shown in supplementary Fig 2. Thanks for the comments and suggestions.

Point 5: In Figure 6C, false positive signals of ZO-1, which should be expressed specifically on cell surface, were shown in nuclei and the changes of the signals by NAS, 4-HNE and Atra was also shown remarkably in nuclei when compared with cell surface. Figure should be replaced

Response:

Thanks for the suggestions. We have repeated this experiment for several times independently using antibodies from different biological company, such as Cell Signaling Technology (CST) and Abcam. We are sorry for not replacing this figure in this version.

Point 6: It is interesting study but there is few new finding when compared with previous published papers. For example, <Neuroprotective action of N-acetyl serotonin in oxidative stress-induced apoptosis through the activation of both TrkB/CREB/BDNF pathway and Akt/Nrf2/Antioxidant enzyme in neuronal cells; DOI: 10.1016/j.redox.2016.12.034 >, < N-Acetyl-Serotonin Protects HepG2 Cells from Oxidative Stress Injury Induced by Hydrogen Peroxide, doi.org/10.1155/2014/310504> and others. The reviewer recommends that the authors address their new findings and also change the title of this study

Response:

Thanks for the suggestions. Recent studies have shown that NAS protects against peroxidative damage in neurons, erythrocytes, retinal cells, lung epithelial cells, lymphocytes, and enterocytes. However, underlying regulation mechanisms of NAS in the intestinal epithelial cells are unknown. As our previous studies were focused on tryptophan metabolism and we have carried out a lot of work in it. Interestingly, we found that the tryptophan and its metabolites (kynurenine, kynurenic acid, 5- hydroxytryptophan, melatonin, and NAS) play an important role in anti-inflammatory and anti-oxidation, so the regulatory mechanism of different tryptophan metabolites were carried out in vitro. NAS as one of the metabolites of tryptophan, the regulation role on oxidative stress in vitro has drawn our much attention. Therefore, the development of this work will complement our systematic study of tryptophan metabolism, and provide basis for the regulation mechanism of NAS on the intestinal epithelial antioxidation. We are sorry for not changing the title of this study.

Round 2

Reviewer 1 Report

Point 1.   The authors answered as follows: “In our recent study, we found that tryptophan supplementation to basal diet of weanling piglets improved growth performance and intestinal barrier. Further study shown that this effect of tryptophan was associated with enhanced tryptophan metabolites, such as kynurenine, kynurenic acid, 5- hydroxytryptophan, melatonin, and NAS in the intestine. Based on these in vivo studies, we conducted experiments to test effect of anti-oxidative effect NAS in intestinal porcine intestinal cells (IPEC-1), as shown in this study. We have added more sentences in the introduction section on our research hypothesis of this version.” However, this reviewer cannot find the description in the Introduction of the revised manuscript.

Point 2. This reviewer cannot find the added sentence in the introduction of the revised file.

Point 3.  Line 257. The authors described that IPEC-1 cells were incubated with 4-HNE for six hours. Nevertheless, in the legend of Figure 1, the cells were incubated for 2 hours. Which is correct?

Point 6.   This reviewer cannot find a correction on the revised file.

Author Response

Response to Reviewer 1 Comments (Round 2)

Point 1 (Round 2): The authors answered as follows: “In our recent study, we found that tryptophan supplementation to basal diet of weanling piglets improved growth performance and intestinal barrier. Further study shown that this effect of tryptophan was associated with enhanced tryptophan metabolites, such as kynurenine, kynurenic acid, 5- hydroxytryptophan, melatonin, and NAS in the intestine. Based on these in vivo studies, we conducted experiments to test effect of anti-oxidative effect NAS in intestinal porcine intestinal cells (IPEC-1), as shown in this study. We have added more sentences in the introduction section on our research hypothesis of this version.” However, this reviewer cannot find the description in the Introduction of the revised manuscript.

Response (Round 2):

Please see Line 63-69 in this version for details. Thanks!

Point 2 (Round 2): This reviewer cannot find the added sentence in the introduction of the revised file.

Response (Round 2):

Please see Line 73-76 in this version for details. Thanks!

Point 3 (Round 2): Line 257. The authors described that IPEC-1 cells were incubated with 4-HNE for six hours. Nevertheless, in the legend of Figure 1, the cells were incubated for 2 hours. Which is correct?

Response (Round 2):

Sorry for the error, it should be two hours. We have corrected it and please see Line 283-286 in this version for details. Thanks!

Point 6 (Round 2): This reviewer cannot find a correction on the revised file.

Response (Round 2):

Thanks for the suggestions. We have corrected the word to make it more accurate. Please see Line 204, Line 247, Line 250, and Line 269 in this version for details. Thanks!

Point 1 (Round 1): The authors studied mechanisms on how N-acetyl serotonin (NAS) could protect epithelial cells from oxidative stress. Nrf2 is a well-known transcriptional factor, and the signaling is a representative of antioxidative defense systems. The authors showed that the protective effect against 4-hydroxy-nonenal (4-HNE) of NAS also was via the Nrf2 signaling. This research should fit the scope of “Antioxidants.” However, the group of Dr. Wu had already published similar data (reference No. 30) using the same epithelial cells with 4-HNE and N-acetylcysteine (NAC). The authors should explain the significance of this study well to convince readers. Otherwise, this study looks a mimic of reference No. 30, where NAC was used instead of NAS. Does the hypothesis proposed in the reference No. 30 have to be independent of the findings produced with NAS in this study? Unless the authors discuss enough, including their previous data, the scientific significance of this paper sounds just like that NAS was used instead of NAC.

Response (Round 1):

Thanks for the comments. We agree that both our previous study (reference No. 30) and the present one address the anti-oxidative damage using IPEC-1 cells. In our previous study, NAC was used and its underlying mechanisms was investigated. We found that NAC protected apoptosis by promote GSH synthesis and MKP-1 mediated inhibition on ERK1/2 signalling. Even though anti-oxidative effect of NAC is well known, our study uncovered a underlying mechanisms for this beneficial effect.

     Our lab is mainly focus nutritional biochemistry study with an aim to improved intestinal barrier and health using piglet as a model. In our recent study, we found that tryptophan supplementation to basal diet of weanling piglets improved growth performance and intestinal barrier. Further study shown that this effect of tryptophan was associated with enhanced tryptophan metabolites, such as kynurenine, kynurenic acid, 5- hydroxytryptophan, melatonin, and NAS in the intestine. Based on these in vivo studies, we conducted experiments to test effect of anti-oxidative effect NAS in intestinal porcine intestinal cells (IPEC-1), as shown in this study. We have added more sentences in the introduction section on our research hypothesis of this version. The conclusion of this work provided basis for the regulation mechanism of NAS (tryptophan metabolite) on the intestinal epithelial antioxidation.

Please see Line 63-69 in this version for details. Thanks!

Point 2 (Round 1): Lines 59-61. The authors must explain the reason why the porcine intestinal epithelial cell line was used in this study. If the authors wanted to evaluate the effect of NAS on human cells, Caco-2, HT-29, and T84 cells had to be served as the in vitro models of small and large intestinal epithelia, respectively.

Response (Round 1):

Thanks for the suggestions. We agree that Caco-2, HT-29, and T84 cells are widely used for in vitro study on human intestinal epithelial related studies. We selected IPEC-1 as a model is based on our in vivo study in piglet, as above-mentioned (in Point 1). We hope our in vitro study might helpful to address what we found in piglets. So, we did not used the human cells. We have added one sentence to explain this point in the introduction section of this version.

Please see Line 73-76 in this version for details. Thanks!

Point 3 (Round 1): 1E. Six hours incubation with 4-HNE (20 μM) caused apoptosis only in 23-30% of IPEC-1 cells. This result suggests that 4-HNE is less toxic on the IPEC-1 cells compared to published data of other cells. This reviewer would like to request the authors to describe a discussion about this phenomenon because the properties may affect the universality of this model.

Response (Round 1):

The toxicity of 4-HNE is highly related to the dosage used, exposure time, and cell density. In the previous study, the cells were treatment with 40 μM of 4-HNE and led to 50% cell death. In the present study, the cells were treated with 20 μM of 4-HNE, and we found 30% cell death following 2 h exposure. We have added sentences in the discussion section of the revised manuscript on this point.

Please see Line 283-286 in this version for details.

Thank you very much!

Point 6 (Round 1): Line 218. If the authors would like to refer to integrity, this reviewer recommends that the authors should show the transepithelial electric resistance (TEER).

Response (Round 1):

Thanks for the suggestions. We have replaced it with mucosal barrier to make it more accurate and being supported by our data provided herein. Please see Line 204, Line 247, Line 250, and Line 269 in this version for details. Thanks.

Reviewer 4 Report

No further comments

Author Response

Thank you very much.